# Inactivation of Pol θ and C-NHEJ eliminates off-target integration of exogenous DNA

Alex N. Zelensky[1], Joost Schimmel[2], Hanneke Kool[2], Roland Kanaar[1] & Marcel Tijsterman[2]

Off-target or random integration of exogenous DNA hampers precise genomic engineering and presents a safety risk in clinical gene therapy strategies. Genetic definition of random integration has been lacking for decades. Here, we show that the A-family DNA polymerase θ (Pol θ) promotes random integration, while canonical non-homologous DNA end joining plays a secondary role; cells double deficient for polymerase θ and canonical non-homologous DNA end joining are devoid of any integration events, demonstrating that these two mechanisms define random integration. In contrast, homologous recombination is not reduced in these cells and gene targeting is improved to 100% efficiency. Such complete reversal of integration outcome, from predominately random integration to exclusively gene targeting, provides a rational way forward to improve the efficacy and safety of DNA delivery and gene correction approaches.

[1] Department of Molecular Genetics, Cancer Genomics Netherlands, Erasmus University Medical Centre, Rotterdam, 3000 CA The Netherlands.
[2] Department of Human Genetics, Leiden University Medical Centre, PO Box 9600, Leiden, 2300 RC, The Netherlands. Alex N. Zelensky and Joost Schimmel contributed equally to this work. Correspondence and requests for materials should be addressed to R.K. (email: R.Kanaar@erasmusmc.nl) or to M.T. (email: M.Tijsterman@lumc.nl)

Gene targeting, the precise engineering of genomes through the homologous integration of exogenous DNA, is hampered by the orders of magnitude higher efficiency of untargeted random integration (RI)[1]. RI does not rely on sequence homology, happens with no or little detectable sequence preference, results in insertional mutations, and is hence often referred to as "illegitimate recombination". The mechanism of RI in higher eukaryotes has been enigmatic for decades, because it has evaded rigorous genetic definition. Integration of exogenous DNA at unpredictable positions in the genome can heavily impact on the functioning of the integrated DNA, as well as its genomic environment, and thus presents a safety risk[2–4]. With the advent of CRISPR-Cas9 technology and its seemingly inevitable implementation in clinically relevant gene-correction strategies[5], it becomes all the more important to devise strategies to counteract the potentially detrimental outcomes of RI. Efforts to shift the RI/HR balance of exogenous DNA integration towards HR have been only marginally successful[6, 7]. One reason is the lack of mechanistic understanding of RI relative to HR. From the analysis of integration sites the involvement of DNA topoisomerases has been suggested but not proven[8, 9]. Because a double-stranded break (DSB) in the chromosome is ultimately required to ligate the incoming DNA, and because DSB induction stimulates RI[10–13], it has been hypothesized that homology-independent DSB end joining (EJ) proteins are involved. In mammalian cells EJ activity is primarily mediated by the C-NHEJ pathway involving Ku70-Ku80, DNA-PKcs, and the LigIV–XRCC4 complex[14]. Strikingly in mouse embryonic stem (ES) cells, where Ku proteins are not essential, complete inactivation of C-NHEJ does not result in any increase in gene targeting[15, 16]. A possible explanation is the existence of alternative pathways for EJ. Indeed, when C-NHEJ is suppressed some residual EJ activity remains. This so-called alternative EJ (Alt-EJ) is genetically not well defined, and may be an umbrella term for several distinct auxiliary or context-specific EJ activities. Because the DNA junctions resulting from EJ activities in C-NHEJ impaired cells are characterized by minute stretches of sequence homology, also the term micro-homology-mediated end joining has been used. Proteins implicated in potentially different alternative EJ activities include PARP1[16, 17], Pol θ[18], CtIP[19, 20], the MRN[21] complex, LigI and LigIII[22, 23].

Here we show that the A-family DNA polymerase θ (Pol θ) promotes most of RI events. In the absence of Pol θ, the major DNA double-strand break (DSB) repair pathway, canonical non-homologous DNA end joining (C-NHEJ), mediates residual RI. We detected not a single RI event in cells lacking both Pol θ and C-NHEJ, indicating that if additional pathways of RI exist, their contribution is at least four orders of magnitude lower. Strikingly, we find that the characteristic unfavorable balance between homologously targeted DNA integration and RI in mammalian cells is completely reversed in cells defective for Pol θ and C-NHEJ.

## Results

**Pol θ and C-NHEJ inactivation prevents random integration.** We used CRISPR-Cas9 to inactivate *Polq*, the gene encoding Pol θ, and key C-NHEJ genes *Ku70*, *Ku80* and *LigIV*, and a combination of *Polq Ku70* and *Polq Ku80* in mouse ES cells (Fig. 1a, Supplementary Figs. 1 and 2); two independent knock-out cell lines were generated for each genotype. The observation that single-mutant cells were hypersensitive toward IR validated functional impairment of the encoded proteins. Strikingly, double mutant cells displayed a synergistic increase in IR sensitivity arguing for redundant activities of C-NHEJ and Pol θ on radiation-induced DNA damage (Supplementary Fig. 2). Next, we

measured the ability of these knockout cells to form stable puromycin-resistant colonies upon RI of transfected plasmid DNA encoding a puromycin resistance gene (Fig. 1a, b). Confirming previous work[15, 16], the frequency of transfected DNA integration was unaffected by C-NHEJ inactivation, however, we found it to be severely reduced in *Polq*[−/−] cells (11% of wild-type). Deficiency in nuclear form of the Alt-EJ protein LigIII[22–24] did not result in RI frequency decrease (Supplementary Fig. 3), which may be due to redundancy of LigIII and LigI as was previously observed in manifestations of Alt-EJ[22, 23]. Our observation that loss of C-NHEJ does not at all affect RI slightly deviates from an earlier study reporting a modest decrease in RI upon LIG4 depletion[25], which could be explained by different cell- or growth characteristics for mouse ES cells versus transformed human somatic cells that could affect usage of EJ pathways: mouse ES cells are primarily in S-phase and are thus perhaps more geared up to repair breaks via HR or alt-EJ. We conclude that RI in mouse ES cells is predominantly resulting from Pol θ-mediated repair and refer to this process as TMEJ (for polymerase Theta-Mediated EJ) to distinguish it from C-NHEJ and to acknowledge the notion that Alt-EJ may also encompass Pol θ-independent repair.

This conclusion is supported by the observation that integration sites in Pol θ proficient cells, but not in *Polq*[−/−] cells, frequently resulted in junctions that contained small insertions, which are suggestive of template-based polymerase action primed by the local presence of homologous sequences (Fig. 2 and Supplementary Data 1, 2). Noteworthy, the observed inserts had a striking similarity with those identified at the junctions of translocations that give rise to Ewing Sarcoma[26]. The occasional presence of templated insertions is a signature feature of Pol θ-mediated repair of DSBs[27, 28].

Simultaneous inactivation of Pol θ and Ku70, Ku80 or LigIV resulted in a complete inability of cells to integrate transfected DNA (Fig. 1b). We did not recover a single puromycin-resistant colony in multiple experiments involving more than $1.5 \times 10^8$ *Polq*[−/−] *Ku70*[−/−], $7.5 \times 10^7$ *Polq*[−/−] *Ku80*[−/−] and $2.4 \times 10^8$ *Polq*[−/−] *Lig4*[−/−] cells. No significant differences in transient transfection efficiency, as measured by expression of fluorescent marker 1 day after transfection, was observed between the genotypes (Supplementary Fig. 2b). We conclude that the residual level of RI in *Polq*[−/−] cells results from C-NHEJ.

We confirmed that total suppression of RI in *Polq*[−/−] *Ku70*[−/−] cells was due to loss of Pol θ by complementing *Polq*[−/−] *Ku70*[−/−] cells with a gene encoding Flag-tagged human Pol θ (Fig. 1c). The wild-type form of human Pol θ restored RI capacity, while mutant versions of the protein with abrogated ATPase or polymerase activity did not.

**Pol θ inactivation facilitates gene targeting.** To test whether the absence of Pol θ affected the balance between RI and HR-mediated targeted integration we used our previously established *Rad54-GFP* gene targeting assay[29], which measures the ratio between random (*GFP* negative cells) and targeted integration (*GFP* positive cells) of the construct targeting exon 4 of the *Rad54* locus using FACS (Fig. 3a, b). In *Polq*[−/−] cells we observed a profound shift in the shape of the FACS profile from RI peak towards HR peak. Strikingly, in *Polq*[−/−] *Ku70*[−/−] cells only HR-mediated gene targeting events were detected. The changes in the FACS profiles were associated with a reduction in the total number of integrants (puromycin-resistant cells regardless of GFP status). These results are consistent with a reduction in the efficiency of RI in *Polq*[−/−] and complete inactivation of RI in *Polq*[−/−] *Ku70*[−/−] cells. This outcome also proves that the observed reduction in RI is not resulting from diminished delivery or

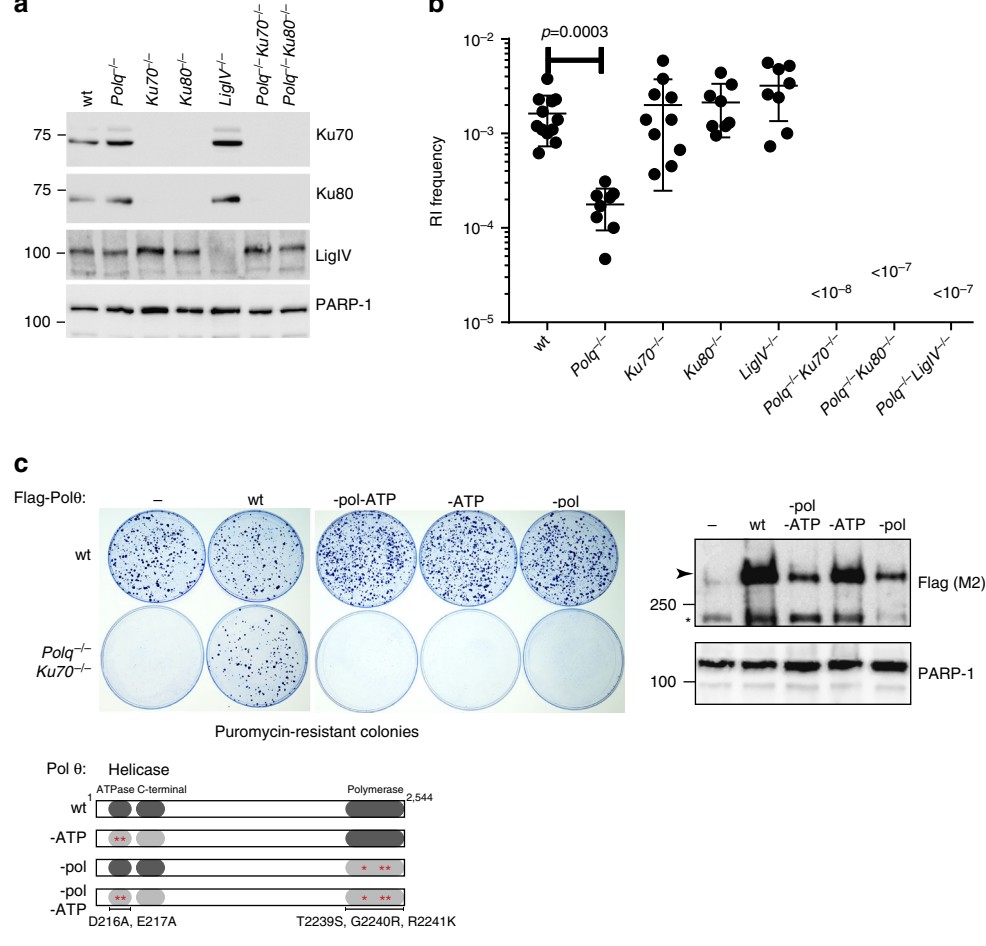

**Fig. 1** RI is reduced in *Polq*⁻/⁻ and completely abrogated in *Polq*⁻/⁻ *Ku70*⁻/⁻, *Polq*⁻/⁻ *Ku80*⁻/⁻ and *Polq*⁻/⁻ *LigIV*⁻/⁻ cells. **a** Immuno-blots confirming the absence of Ku70, Ku80, and LigIV from wild type and *Polq*⁻/⁻ ES cells. **b** Mouse ES cells of indicated genotypes were electroporated with linearized plasmid DNA encoding a puromycin resistance gene and a GFP expression plasmid to measure transfection efficiency. Colony counts were adjusted for transfection and plating efficiency. Means and s.e.m. are plotted. **c** *Polq*⁻/⁻ and wild-type control cells complemented with wild-type and mutant versions of the human POLQ cDNA were transfected with a plasmid encoding a puromycin-resistance gene. After 10 days of puromycin selection CBB-stained plates were photographed. Right-hand panel shows immuno-blots of Pol θ levels in the clones used in the left-hand panel. Arrowhead indicates full-length Flag-Pol θ protein, asterisk indicates non-specific signal

stability of transfected DNA[30]. Since induction of DSBs stimulates RI, we next tested whether γ-irradiation affected the HR/RI ratio. Indeed, we observed significant stimulation of RI in wild-type and single mutant cells (Fig. 3b). However, even under these stimulatory conditions, no RI was detected in *Polq*⁻/⁻ *Ku70*⁻/⁻ cells, and only HR manifested. As most gene targeting strategies currently involve stimulation by DSB induction at the target locus, we tested whether loss of Pol θ and C-NHEJ is also beneficial in the context of CRISPR-Cas9-stimulated gene targeting, and indeed it is (Supplementary Fig. 4).

HR-mediated gene targeting at exon 4 of *Rad54* is very efficient: 30–70% compared to ~5% for an average location in mouse ES cells. Therefore, we tested whether the high gene targeting efficiency we observed in *Polq*⁻/⁻ cells will be maintained in a more typical scenario. Constructs that target exon 9 or 18 of *Rad54* (Fig. 3a) recombine with much lower frequency (<5% in wild type cells, which results in a GFP-positive cell frequency below detection threshold). For these constructs, we also found a profound shift from RI towards HR in *Polq*⁻/⁻ cells (Fig. 3d). This phenomenon is not specific for the *Rad54* locus, as we detected a three-fold increase in relative gene targeting efficiency (ratio between random and targeted integration frequencies, resulting in *puroR+hygroR−* and *puroR+hygroR+* phenotypes, respectively) at

another genomic locus, i.e., the *Pim1* gene (Fig. 3c). We conclude that Pol θ inhibition is a useful method to dramatically and generally increase gene targeting efficiency.

**Pol θ inactivation does not affect homologous recombination.** Recent studies suggest that Pol θ inhibits HR by interacting with Rad51 recombinase[18, 31]. The strong reduction in RI we observed in *Polq*⁻/⁻ cells can account for the increase in the HR/RI ratio measured in the Rad54-GFP gene targeting assay, however an increase in HR efficiency may also contribute. We used a direct-repeat GFP (DR-GFP) assay[32] to measure DSB-induced gene conversion and found no effect from genetically inactivating Pol θ alone (Fig. 4a). In addition, we found no significant difference in the absolute frequency of targeted integration of the *Pim1* targeting construct in Pol θ proficient and deficient mouse ES cells ($p = 0.37$, $n = 4$). However, in the DR-GFP assay, the simultaneous inactivation of Pol θ and Ku80 resulted in a higher yield of gene conversion events (Fig. 4a). Thus, a potential increase in HR through the lack of Pol θ is unlikely to contribute to the increased gene targeting efficiency in *Polq*⁻/⁻ mouse ES cells, but may be a factor when both C-NHEJ and TMEJ are inactive, as these end joining factors may locally act redundantly to suppress HR.

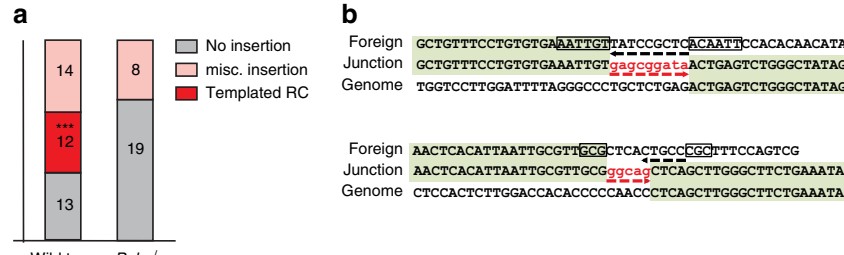

**Fig. 2** Different junctional diversity at RI sites in wild-type and $Polq^{-/-}$ cells. **a** Schematic representation of junctions found at RI sites: (i) simple deletions that are without insertions (*in grey*), (ii) complex junctions that contain insertions, which display sequence identity to the transfected DNA but in a reverse complement (RC) orientation (*in red*), and (iii) complex junctions with untraceable miscellaneous insertions (*in pink*); *** denotes $p < 0.001$ in Fischer's exact test. **b** Two representative cases of insertion-containing transfected DNA-genome junctions. Junctional sequences that are identical to the transfected (foreign) DNA (*in bold*) or the mouse genome are marked. Micro-homologous sequences are boxed. Identical sequences in a RC orientation are underlined

**Pharmacological manipulation to improve gene targeting**. Our results described above imply that transient inactivation of RI by small molecule inhibition of TMEJ and C-NHEJ could obviate the need for DSB induction (and thereby avoid its associated side effects) for precise gene targeting in certain genomic engineering scenarios. We tested whether suppressing residual RI activity in $Polq^{-/-}$ cells by NU-7441, a compound implicated in C-NHEJ inhibition by targeting DNA-PKcs activity, increases gene targeting efficiency at *Rad54* exons 9 and 18 (Fig. 4b). Indeed, treatment for 24 h after transfection with 10 μM of NU-7441 resulted in a significant increase in the ratio of HR to RI in $Polq^{-/-}$ but not wild-type cells. Thus, our results revealed the essential targets, Pol θ and C-NHEJ, and demonstrate that pharmacological manipulation can provide an opportunity to enhance the efficiency of precise genomic engineering.

## Discussion

We established the up until now elusive genetic dependencies of RI in mammalian cells. We demonstrate that exactly two genome maintenance pathways, TMEJ and C-NHEJ, carry out all detectable RI, with TMEJ being the default dominant pathway. Similar observations have now been made in human cells[33]. It has been known for four decades that transforming cells with exogenous DNA leads to integration at unpredictable positions in the genome[34, 35], which hampers clinically relevant gene-correction strategies. The genetic definition of the major promoters of RI of exogenous DNA presented here allows the rational design of methods to improve the efficacy and safety of gene correction approaches for industrial or medical purposes.

## Methods

**Cell lines**. Mouse ES cells were routinely cultured on gelatinized (0.1% gelatin in water) dishes in media comprising 1:1 mixture of DMEM (Lonza BioWhittaker Cat. BE12-604F/U1, with Ultraglutamine 1, 4.5 g/l Glucose) and BRL-conditioned DMEM, supplemented with 1000 U/ml leukemia inhibitory factor, 10% FCS, 1× NEAA, 200 U/ml penicillin, 200 μg/ml streptomycin, 89 μM β-mercaptoethanol. The parental wild-type line (IB10) was a specific pathogen free clonal isolate of the 129/Ola E14 line[36].

For the generation of *Polq*, *Ku80* (*Xrcc5*), and *LigIV* knockout cell-lines, ES cells were cultured on gelatin-coated plates containing irradiated primary mouse embryonic fibroblasts in knockout Dulbecco's Modified Eagle Medium (Gibco) supplemented with 100 U/ml penicillin, 100 μg/ml streptomycin, 2 mM GlutaMAX, 1 mM sodium pyruvate, 1× non-essential amino acids, 100 μM β-mercaptoethanol (all from Gibco), 10% FCS and 1000 U/ml leukemia inhibitory factor. IB10 cells were transfected with plasmids expressing Cas9-WT-2A-GFP (pX458 and sgRNAs targeting different sequences in the three genes using Lipofectamine 2000 (Invitrogen)). Cells were seeded at low density and maintained with regular medium changes for 8–10 days until colonies were formed. Colonies were picked and grown in 96-wells format, at (semi) confluence cells were split to two 96-wells plates. One plate was used for sub-culturing of cells; the other plate was used for DNA isolation and PCR analysis of the individual clones. Restriction fragment length polymorphism (RFLP, based on the loss of a unique restriction

site) of PCR-products was used to identify clones with a bi-allelic mutation. PCR-products of clones that showed bi-allelic mutation were sent for sequencing to confirm the introduction of a premature stop-codon due to deletions/insertions that cause a frameshift (Supplementary Fig. 1b). *Polq Ku80* double knockout cell lines were generated by targeting the *Polq* gene in the resulting Ku80 deficient cell lines. Experiments described in this study have been done using at least two independent clonally derived knockout lines per gene. An overview of the targeted sequences and the oligonucleotides and restriction sites used for the RFLP analysis can be found in Supplementary Data 3.

*Ku70* (*Xrcc6*) knockout ES cells were generated by CRISPR-Cas9 stimulated gene targeting with a homologous template. The sgRNA recognizing the sequence in exon 4 of *Ku70* was cloned in pX459 vector[37]. The information on oligonucleotides and constructs is provided in Supplementary Data 3 and 4. The donor template was produced by PCR-amplification of two homology arms from IB10 genomic DNA (3.3 and 2.3-kb long), the PGK-hygro selection cassette and the high-copy plasmid backbone pBluescript, and ligated using the Gibson isothermal assembly method[38]. The construct was partially sequenced to verify the junctions between the selection cassette and homology arms. Oligonucleotides used in cloning and genotyping are listed in Supplementary Data 3 and 4. Homologous template and sgRNA+Cas9 expression constructs were co-electroporated into $1–2\times10^7$ IB10 or $Polq^{-/-}$ cells. Selection with 200 μg/ml Hygromycin B (Roche) was started 1 day after electroporation and maintained with regular media changes for 8–10 days until colonies formed. Colonies were picked using a sterile pipette tip under the microscope, dispersed with Trypsin-EDTA, divided for re-plating and for lysis (50 mM KCl, 10 mM Tris–HCl pH 9, 0.1% Triton-X100, 0.15 mg/ml proteinase K, 60 °C 1 h, 95 °C 15 min) followed by genotyping PCR with K70-e34-scrF, K70-e34-scrR and PGK-R1 primers. The absence of the band expected form the wild-type allele was interpreted as an indication of the bi-allelic modification of the locus (~50% clones). A subset of them was expanded, re-genotyped and tested by immuno-blotting.

**Immuno-blotting**. Whole cell lysates were fractionated by Tris-glycine SDS–PAGE, transferred to nitrocellulose membrane and blotted (5% milk, 0.05% Tween 20 PBS, 4 °C overnight) with the following antibodies: Ku70 (C-19 goat polyclonal, 1:2500, Santa Cruz sc-1486), Ku80 (M-20, 1:2000 Santa Cruz sc-1485), LigIV (1:2000 Abcam ab80514), Flag tag (M2 mAb, 1:5000, Sigma), PARP-1 (C2-10 mAb, 1:5000, ENZO). Uncropped scans of the immuno-blots are show in Supplementary Fig. 5.

**Clonogenic survivals**. Cells were plated in triplicates in 6-well plates at 100–4000 per well and irradiated using $^{137}$Cs (*Ku70*) or YXlon X-ray generator (*Ku80*) immediately after seeding. Colonies were allowed to form for 7 days, after which cells were fixed and stained.

**DNA constructs**. pLPL was derived from the construct loxP-PGK-gb2-neo-polyA-loxP cassette in pGEM-T Easy (a gift from Francis Stewart[39]) by replacing neomycin phosphotransferase with puromycin N-acetyltransferase CDS. Gene targeting constructs for Rad54 exons 9 and 18 were produced by recombineering with mobile agents[39] with a cassette containing PCR-amplified fragment of the previously described exon 4 gene targeting vector[40], encoding part of the RAD54 CDS downstream of the targeted exons, GFP CDS and PGK polyadenylation signal, and the loxP-flanged PGK-EM7-neo dual selection cassette into a BAC bMQ-422M23 Sanger 129/Sv BAC library[41]. Fragments of the modified BAC were subcloned into high-copy pBluescript vector using recombineering (retrieval). Plasmids SpCas9 (BB)-2A-GFP (PX458) and SpCas9(BB)-2A-puro (PX459 v2.0) were a gift from Feng Zhang (Addgene plasmid #48138, #62988). A plasmid based on the pENTR233 vector and encoding human Pol θ was purchased from the ORFeome collaboration collection (Dharmacon), and re-cloned into pAZ125 PiggyBac vector,

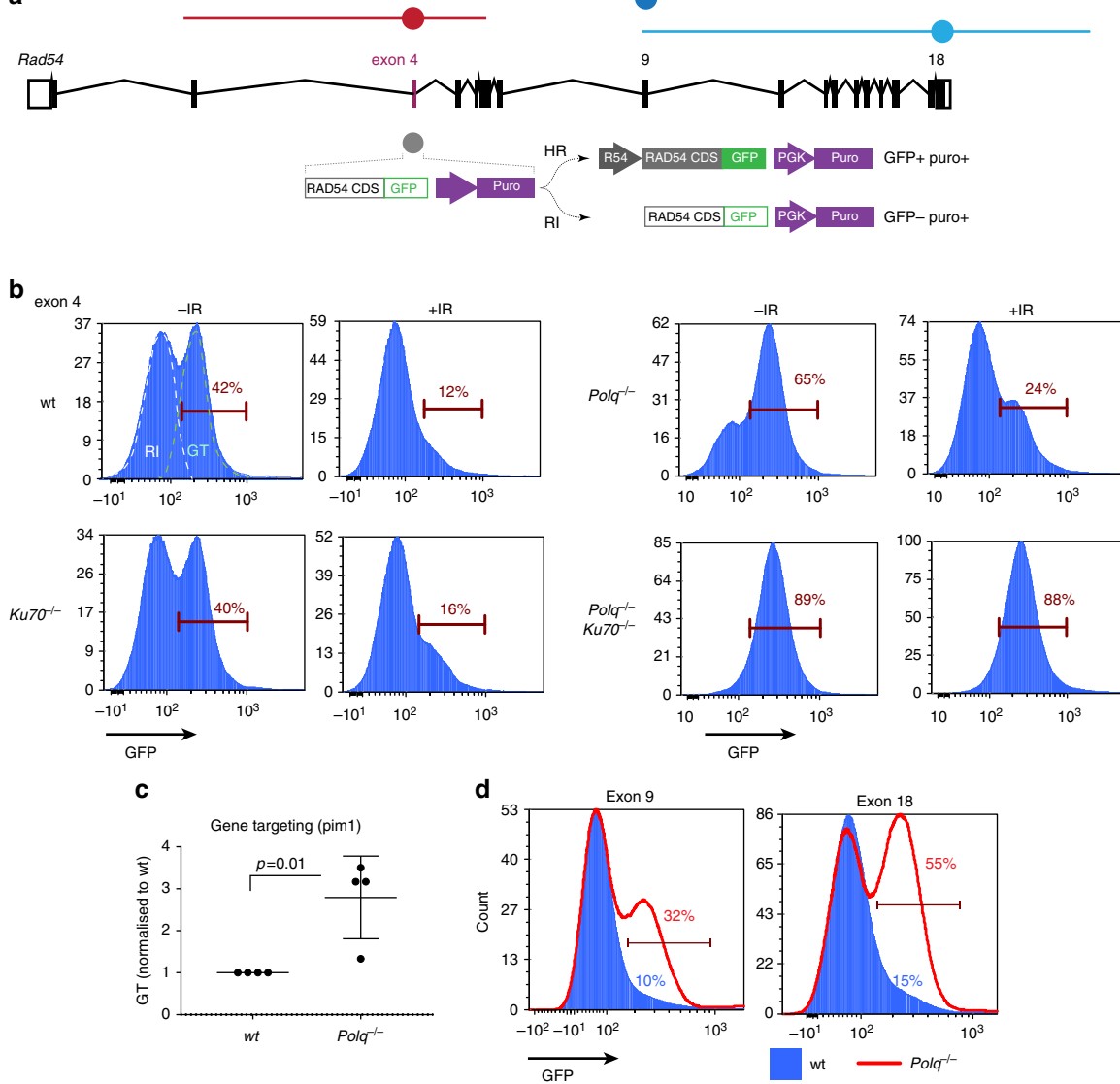

**Fig. 3** Highly efficient gene targeting in the absence of Pol θ. **a** Map of the *Rad54* locus showing regions of homology with the gene targeting constructs used in the study. The oval on the construct map indicates the location of the cassette containing the downstream part of the Rad54 CDS fused to GFP, and a selection cassette. **b** FACS frequency histograms from the Rad54-GFP gene targeting assay in unirradiated and irradiated (200 mGy) cells. Marker used to determine the percentage of Rad54-GFP-positive cells is shown. As the shoulders of RI and GT peaks overlap, the number of events within the marker under-reports or over-reports the true fraction of targeted cells, when the distribution is strongly skewed toward GT or RI, respectively. **c** Gene targeting efficiency at the *Pim1* locus (means and s.e.m. are plotted). **d** Gene targeting efficiency of exon 9 and exon 18 Rad54-GFP knock-in constructs in wild type (filled histograms) and *Polq⁻/⁻* (red outline) cells

which was derived from 5′-PTK-3′[42] by replacing the transposon cargo (XhoI-SpeI fragment) with two expression cassettes: loxP-flanked PGK→neo and CAG→[ClaI XhoI]~bGHpa (bovine growth hormone polyadenylation signal). BglII-KasI fragment of Pol θ CDS and two PCR-amplified fragments covering parts of Pol θ CDS not included in the BglII-KasI fragment were inserted into pAZ125 using Gibson assembly[38]. Point mutations to produce D216A, E217A (ATPase-deficient), T2239S, G2240R, R2241K (polymerase-deficient), and the double ATPase+polymerase deficient forms of Pol θ were introduced by PCR amplification with Q5 polymerase (NEB) of the POLQ CDS fragments, and insertion into pAZ125 using Gibson assembly. Two sgRNAs targeting the *Rad54* exon 18 region between the homology arms of the corresponding targeting construct, and thus replaced by the GFP-PKG-neo cassette upon recombination, were cloned in pX459.

**Gene targeting assays**. For *Rad54* gene targeting assay 10⁷ were electroporated with 10 μg PvuI-linearized gene targeting construct and plated in 10 gelatinized 10 cm dishes. For experiments involving irradiation, cells were seeded in duplicate, one of which irradiated with 200 mGy using ¹³⁷Cs source. The following day media was exchanged to selective media (1.5 μg/ml puromycin for exon 4

constructs, 500 μg/ml G418 for exon 9 and 18 constructs). Where indicated, sgRNA+Cas9 expression construct was co-transfected with the gene targeting construct, which in these experiments was not linearized. After 8–10 days of selection, cells were trypsinized, collected by centrifugation, fixed by resuspending in 1 ml of 1% paraformaldehyde in PBS, incubated for 15 min and mixed with an equal volume of 0.2% Triton X100 solution in PBS to enhance the separation between GFP+ and GFP− peaks. FACS analysis was performed using BD LSR Fortessa instrument. *Pim1* gene targeting assay[32] was performed using the DR-GFP reporter construct which carries a hygromycin resistance gene that can be expressed from the *Pim1* promoter upon targeted integration and a puromycin resistance gene that is under PGK promoter and is expressed after both targeted and random integration. After electroporation cells were divided into two dishes: 10% for puromycin selection (random and targeted integration) and 90% for hygromycin selection (targeted integration). An aliquot was taken to estimate plating efficiency without selection. After 8–10 days of selection colonies were stained and counted. The ratio of hygromycin-resistant to puromycin-resistant colonies was interpreted as relative gene targeting efficiency, while the number of hygromycin-resistant colonies per viable plated cell was defined as absolute targeting frequency.

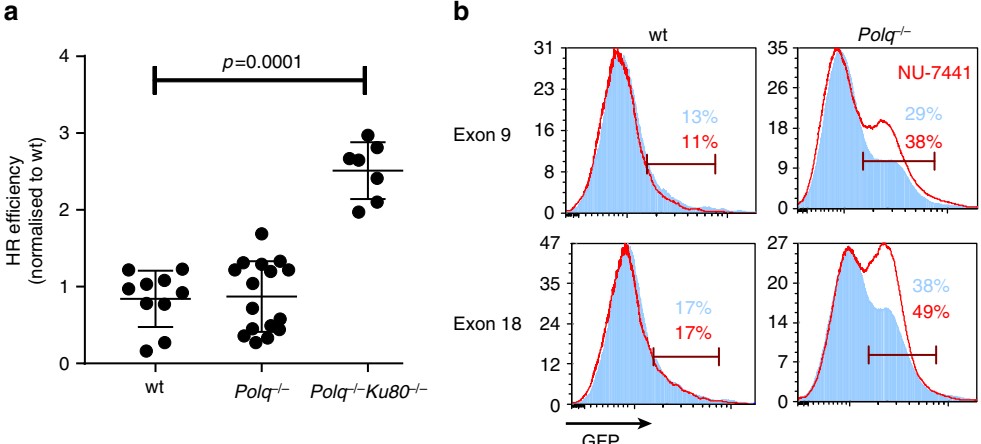

**Fig. 4** HR frequency and pharmacological manipulation of HR/RI ratio in *Polq⁻/⁻* cells. **a** DR-GFP reporter constructs were integrated in wild-type and mutant cell lines at the *Pim1* locus. Reporter-containing cells were co-lipofected with I-SceI expression construct and RFP expression plasmid. Gene conversion frequency (GFP+) and transfection efficiency (RFP+) was measured by FACS 48 h after transfection. Means and s.e.m. are plotted. **b** Wild-type (*left*) and *Polq⁻/⁻* cells were electroporated with exon 9 or exon 18 Rad54-GFP gene targeting constructs and 10 μM NU-7441 (*red outline*) was added to the media immediately after electroporation and removed 24 h later. The HR/RI ratio was determined by FACS

**Random integration assay**. Fifteen million cells were electroporated (118 V, 1200 μF, ∞ Ω, exponential decay, GenePulser Xcell apparatus (BioRad), 2 mm cuvette (BTX)) with 15 μg linearized (DraI digestion, followed by phenol extraction and ethanol precipitation) or circular pLPL plasmid containing puromycin N-acetyltransferase under mouse PGK promotor and 15 μl of circular GFP reporter plasmid (pEGFP-N1). For mutant cell lines electroporations were done in duplicates or triplicates to compensate for low plating efficiency. Cells were re-suspended in 5.5 ml media and seeded at 1/5 and 4/5 dilution into 10 cm dishes; 2–4 μl samples of the 1/5 dish were plated in triplicates for plating efficiency estimation; the remainder of the suspension was seeded into a 24-well plate and analyzed by FACS the following day to estimate transfection efficiency. Puromycin selection (1.5 μg/ml) was started the following day and maintained for 8–10 days. Colonies were stained (0.25% Coomassie Brilliant Blue R (CBB), 40% methanol, 10% acetic acid), photographed and counted manually using FIJI CellCounter plugin or with OpenCFU[43].

**Mapping integration sites**. Insertion junctions were amplified and analyzed as described previously[44]. Briefly, cells were transfected with linear pLPL plasmid PCR-products using Lipofectamine 2000 (Invitrogen). Puromycin (1.5 μg/ml) was added 24 h after transfection and colonies were allowed to grow for 8–10 days. Individual puromycin-resistant clones were isolated and grown on 96-wells plates. At confluency, DNA was isolated from individual wells and plasmid-genome junctions were amplified via TAIL-PCR[45]. Sequences of plasmid-specific and arbitrary degenerate (AD) oligonucleotides used for TAIL-PCR can be found in Supplementary Data 4. Sanger sequencing was performed using the innermost TAIL-PCR oligonucleotide (pLPL_For-3). Subsequent automated analysis was performed by custom scripts. High-quality DNA stretches (base error quality <0.1, stretch ≥50 nucleotides) were identified within each Sanger sequence and further analyzed. Blast against the mouse genome and the pLPL plasmid PCR-product was performed to identify junctions. For each junction, the type of junction was determined (plasmid product–plasmid product or plasmid product–mouse genome) and whether overlap existed between the two top BLAST hits. If the two top BLAST hits overlapped it indicated the use of (micro)homology. When no overlap between the two top BLAST hits was found, the junction was annotated as a filler. When fillers were large enough (≥5 nucleotides) the origin of the filler was searched for in the pLPL plasmid and in the mouse genome at the location the junction mapped. Total of 100 base pairs at both sides of the mapped location were searched in normal and reverse complement orientation using the longest common substring approach[44].

**Data Availability**. All data generated or analyzed during this study is included in this published article and its supplementary information.

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

## Acknowledgements

We thank Robin van Schendel for bio-informatics support. Lig3 deficient cells were kindly provided by Maria Jasin. The work in R.K.'s laboratory was funded by the Cancer Genomics Netherlands Gravitation Program and an ECHO grant from NWO Chemical Sciences. J.S. is funded by a VENI grant from NWO-Life Sciences.

## Author contributions

A.N.Z. and J.S. designed and performed experiments, interpreted results and wrote the manuscript; H.K. performed experiments; R.K. and M.T. interpreted results and wrote the manuscript.

## Additional information

**Competing interests:** The authors declare no competing financial interests.

