## [Peer Review file · Nature Communications]

Reviewers' comments:

Reviewer #1 (Remarks to the Author):

In this manuscript by Zelensky et al. the authors report on an interesting role for PolQ and C-NHEJ in random integration (RI). The authors analyze mammalian cells lacking PolQ and Ku70/80 (and Lig4) and show that RI is completely abrogated when MMEJ and NHEJ are blocked. The authors also conclude that the decrease in RI translates into enhanced gene targeting.

This reviewer finds this study to be interesting and novel. The experiments are well controlled and that data presented is clear. The observations provide a good start, but additional mechanistically understanding is necessary. Specifically, it is important to provide to decipher the basis of how PolQ promote RI. Are RI events taking place during S phase, and linked to the role of PolQ during replication. Alternatively, is this function of PolQ during RI linked to its role during alt-NHEJ, in which case, blocking Lig3 (nuclear) or CtIP should have the same effect?

In addition, I have a number of concerns related to the inconsistencies between the data presented in this manuscript and previous literature, and experiments reported in the co-submitted manuscript.

a) Very nice work from the labs of Wood and Mckenna and Gilles, show that PolQ is a radiosensitizer (Yousefzadeh Plos genetics Fig 1, and Higgins et al., Figure 3). Work y Ceccaldi et al., and Mateos Gomez et al., show that depletion of PolQ increase the frequency of HR. The knockout generated by the authors does not seem to fully recapitulate both phenotypes, which casts some doubts as to whether the Cell line is a complete knockout. The deletion of the gene in the close was not confirmed by western. For this reason, it is important to confirm the observation in an independent PolQ-KO cell line.

b) In figure 1, Zelensky et al., show that depletion of Lig 4, Ku 70, and Ku80 have no effect on the frequency of RI. In contrast, Saito et al., show a significant (2 fold) reduction in RI events when Lig4 is depleted. This contradiction need to be addressed prior to publication.

Reviewer #2 (Remarks to the Author):

The genetic determinants of illegitimate gene targeting in mammals, also referred to as random integration (RI) of targeting vectors, have not been fully understood. Attempts to decrease RI via knockout of classical NHEJ proteins have been unsuccessful. In this paper, Zelensky and colleagues show that the majority of RI proceeds through a polymerase theta (Pol θ)-mediated alternative end joining pathway. Furthermore, RI is completely abolished in the absence of both c-NHEJ and theta-mediated end joining (TMEJ). One of the most interesting aspects of this study is that treatment of POLQ $-/-$ mouse ES cells with a DNA-PKcs inhibitor results in increased gene targeting efficiencies. In combination, the results are novel will be very useful for investigators attempting gene targeting experiments in mice. However, a few additional experiments and clarifications are needed to maximize the impact of the study.

Major points:

1. The authors show that mutation of Pol θ increases accurate gene targeting even in cells exposed to IR. Given that most current gene targeting strategies would use CRISPR-Cas9 to induce breaks at the desired target locus, they should also show that loss of Pol θ and c-NHEJ results in 100% (or greatly increased) gene targeting accuracy in the context of a CRISPR-induced break in one of their Rad54 gene targeting experiments.

2. None of the RI junctions appear to involve annealing at microhomologous sequences. This is unusual, given that Pol θ has been shown to also mediate simple MMEJ. Were MH junctions not

observed or just not characterized as such?

3. Statistical analysis is needed for the junction classification in Figure 2.

4. The supplemental Excel file with RI junctions should be converted into a table that can be included as part of a supplementary table, rather than just as an Excel file, so that readers can easily inspect the RI junctions.

5. Similarly, were any RI junctions characterized for the Rad54 or Pim1 Polq $-/-$ mutant gene targeting experiments? Inclusion of these data would provide further support for the role of Pol θ in alt-EJ mediated RI.

Minor points:

6. Pol θ rescue constructs with mutations in the ATPase or polymerase domains failed to restore RI capacity, while wild-type Pol θ did complement. Why would ATPase activity be needed for RI? The authors may wish to speculate on this.

7. Figure 4A and page 5 lines 15-16: Although no effect on HR in the DR-GFP assay was observed in the Polq $-/-$ cells, there was an effect in the absence of both Pol θ and Ku80. Therefore, the increased gene targeting efficiency in Polq $-/-$ Ku80 $-/-$ mouse ES cells might actually be due to a stimulatory effect on HR.

8. Page 5 line 16: What is meant by "absolute frequency of gene targeting" at the pim1 locus? How does this relate to the 3-fold increase in relative gene targeting efficiency seen in Figure 3c?

9. Figure 2 - A diagram showing the mechanism by which Pol θ might generate the two RI junctions would be helpful (indicating the role of the reverse complement sequences). This could be included in supplemental information.

10. Figure 4b - The values for RI and gene targeting should be indicated as in Figure 3.

11. The references have been separated between references 31 and 32.

Reviewer #3 (Remarks to the Author):

This ms. describes a well-designed series of experiments showing that pol theta play an important role in RI in mouse cells. The experiments take advantage of their experience with mouse cells for gene targeting and show how RI is regulated in these cell lines. These results will be highly cited.

Point to point response Zelensky et al.,

We would like to thank all reviewers for their time and support, and for the constructive remarks and suggestions, which in our opinion helped us to increase the quality and readability of our manuscript.

Reviewer #1

In this manuscript by Zelensky et al. the authors report on an interesting role for PolQ and C-NHEJ in random integration (RI). The authors analyze mammalian cells lacking PolQ and Ku70/80 (and Lig4) and show that RI is completely abrogated when MMEJ and NHEJ are blocked. The authors also conclude that the decrease in RI translates into enhanced gene targeting.

This reviewer finds this study to be interesting and novel. The experiments are well controlled and that data presented is clear.

We thank the reviewer for his/her positive evaluation and kind words

The observations provide a good start, but additional mechanistically understanding is necessary. Specifically, it is important to provide to decipher the basis of how PolQ promote RI. Are RI events taking place during S phase, and linked to the role of PolQ during replication. Alternatively, is this function of PolQ during RI linked to its role during alt-NHEJ, in which case, blocking Lig3 (nuclear) or CtIP should have the same effect?

We now include a set of experiments in which we addressed the requirement for Lig3 in random integration of transfected DNA. We obtained mouse ES cells from the laboratory of Maria Jasin that are either wild type or mutated for Lig3, yet expressing mitochondrial Lig3 or Lig1 to avoid cellular lethality (as described in Simsek et al., Nature 2011). We found that cells lacking nuclear Lig3 had no defect in random integration (Supplemental Fig. 3).

While this observation may lead to the notion that random integration by Pol theta is not alternative end-joining, we favor a broader view of the ill-defined Alt-EJ pathway which in certain contexts and/or cell-cycle stages may depend on Lig3, yet in other contexts/stages can employ both Lig3 and Lig1, explaining the observed variation in genetic requirement for different EJ activities.

We consider it technically impossible within the available timeframe to address cell-cycle dependency of RI in mouse ES cells given that i) mouse ES cell populations are predominantly S-phase cells and spend less than 15% of their ~ 12 hour cell cycle in G1, ii) the unavailability of robust synchronization protocols, and iii) most importantly the persistence of transfected DNA (which based on our unpublished work is at least 4 and up to 24 hours) will complicate the interpretation experiments with synchronized cells.

In addition, I have a number of concerns related to the inconsistencies between the

data presented in this manuscript and previous literature, and experiments reported in the co-submitted manuscript.

a) Very nice work from the labs of Wood and Mckenna and Gilles, show that PolQ is a radiosensitizer (Yousefzadeh Plos genetics Fig 1, and Higgins et al., Figure 3). Work y Ceccaldi et al., and Mateos Gomez et al., show that depletion of PolQ increase the frequency of HR. The knockout generated by the authors does not seem to fully recapitulate both phenotypes, which casts some doubts as to whether the Cell line is a complete knockout. The deletion of the gene in the close was not confirmed by western. For this reason, it is important to confirm the observation in an independent PolQ-KO cell line.

We apologize for not being clearer in the initial submission. All our data were derived from two different independently generated Polq knockout alleles that behaved identical in all assays analyzed (including hypersensitivity towards ionizing radiation Supplemental Fig 2A). This is now made clearer in this version of the manuscript. Furthermore, we have added new data where we show that PolQ depletion in LigIV knockouts also results in abolished RI. These cell lines are independently generated (for Polq knockout, see new version of Supplemental Fig. 1).

b) In figure 1, Zelensky et al., show that depletion of Lig 4, Ku 70, and Ku80 have no effect on the frequency of RI. In contrast, Saito et al., show a significant (2 fold) reduction in RI events when Lig4 is depleted. This contradiction need to be addressed prior to publication.

We now include a possible explanation for this discrepancy in the text of the manuscript. We suggest that the observed differences may very well result from different cell- or growth characteristics for mouse ES cells versus transformed human somatic cells that may influence usage of EJ pathways. More specifically, and as mentioned above, mouse ES cells are primarily in S-phase and thus perhaps more geared up to repair (replication associated) breaks via HR and TMEJ.

Reviewer #2 (Remarks to the Author):

The genetic determinants of illegitimate gene targeting in mammals, also referred to as random integration (RI) of targeting vectors, have not been fully understood. Attempts to decrease RI via knockout of classical NHEJ proteins have been unsuccessful. In this paper, Zelensky and colleagues show that the majority of RI proceeds through a polymerase theta (Pol θ)-mediated alternative end joining pathway. Furthermore, RI is completely abolished in the absence of both c-NHEJ and theta-mediated end joining (TMEJ). One of the most interesting aspects of this study is that treatment of POLQ $-/-$ mouse ES cells with a DNA-PKcs inhibitor results in increased gene targeting efficiencies. In combination, the results are novel will be very useful for investigators attempting

gene targeting experiments in mice. However, a few additional experiments and clarifications are needed to maximize the impact of the study.

We thank the reviewer for his/her positive evaluation and suggestions to improve the manuscript.

Major points:

1. The authors show that mutation of Pol θ increases accurate gene targeting even in cells exposed to IR. Given that most current gene targeting strategies would use CRISPR-Cas9 to induce breaks at the desired target locus, they should also show that loss of Pol θ and c-NHEJ results in 100% (or greatly increased) gene targeting accuracy in the context of a CRISPR-induced break in one of their Rad54 gene targeting experiments.

We have performed CRISPR-induced gene targeting in wild type, Polq^{-/-}, Ku70^{-/-} and Polq^{-/-} Ku70^{-/-} cells and now include the data in Supplementary Fig. 4. We demonstrate a profound shift from random integration to gene targeting already in wildtype cells. The residual random integration that is still noticeable in wildtype is completely absent in Polq^{-/-} Ku70^{-/-}. This data also reinforces the point that gene targeting, also stimulated by DSBs, does not require Pol theta.

2. None of the RI junctions appear to involve annealing at microhomologous sequences. This is unusual, given that Pol θ has been shown to also mediate simple MMEJ. Were MH junctions not observed or just not characterized as such?

We realize that the usage of homology is indeed not easily deduced from the data we provided. We now correct this by including a table that graphically represents all junctions between the exogenous DNA and the mouse genome (that are without inserts). We find that micro-homology usage is elevated as compared to what would be expected by chance, however, we surprisingly found that this was also true for junctions in *polq*^{-/-} cells, which we demonstrate to be cNHEJ dependent. At this stage we do not have an explanation for this observation.

The technology to map integrations proved to be very difficult. We used TAIL PCR-based approaches and splinkerette technology, both of which are terribly inefficient. We are currently developing new methods that also involve NGS approaches to increase sample size, robustness and throughput while reducing labor (which is now troublesome). Only in a small percentage of RI clones we were able to identify the site of integration using conventional (sub-optimal) techniques.

3. Statistical analysis is needed for the junction classification in Figure 2.

This is now included.

4. The supplemental Excel file with RI junctions should be converted into a table that

can be included as part of a supplementary table, rather than just as an Excel file, so that readers can easily inspect the RI junctions.

This has now been done

5. Similarly, were any RI junctions characterized for the Rad54 or Pim1 Polq $-/-$ mutant gene targeting experiments? Inclusion of these data would provide further support for the role of Pol θ in alt-EJ mediated RI.

These have not been characterized: detection of junction between randomly integrated gene targeting construct is particularly challenging (if at all possible) due to the presence of large fragments of genomic DNA in the construct (homology arms) flanking the transgene-specific sequence (selection cassette) to which mapping primers can be anchored.

Minor points:

6. Pol θ rescue constructs with mutations in the ATPase or polymerase domains failed to restore RI capacity, while wild-type Pol θ did complement. Why would ATPase activity be needed for RI? The authors may wish to speculate on this.

We feel uncomfortable speculating on this issue, as frankly we have really do not know at this point. Our unpublished work in *C. elegans* shows that the helicase domain is required for all phenotypes associated with TMEJ, while *Drosophila* work presented at conferences has pointed towards specialized functions of the ATP activity. This work is yet unpublished.

7. Figure 4A and page 5 lines 15-16: Although no effect on HR in the DR-GFP assay was observed in the Polq $-/-$ cells, there was an effect in the absence of both Pol θ and Ku80. Therefore, the increased gene targeting efficiency in Polq $-/-$ Ku80 $-/-$ mouse ES cells might actually be due to a stimulatory effect on HR.

Indeed, this cannot be ruled out and we thank the reviewer for pointing this out. The concluding sentence of the corresponding paragraph was modified to mention this possibility.

8. Page 5 line 16: What is meant by “absolute frequency of gene targeting” at the pim1 locus? How does this relate to the 3-fold increase in relative gene targeting efficiency seen in Figure 3c?

Absolute frequency is defined as the number of targeted integration events (puroR+hygroR+) per viable transfected cells, while relative targeting efficiency is defined as the fraction of targeted integrants among all integrants. Thus a 3-fold increase in relative gene targeting efficiency due to reduced random integration is possible without a corresponding increase in absolute frequency of targeted integration.

We amended the text in the results and methods sections to clarify this.

9. Figure 2 - A diagram showing the mechanism by which Pol θ might generate the two RI junctions would be helpful (indicating the role of the reverse complement sequences). This could be included in supplemental information.

We now refer to a paper published by Zucman-Rossi (Proc. Natl. Acad. Sci USA, 1988), which reports similar junctions in Ewing Sarcoma tumor cells and also generated a schematic diagram to explain their origin. Because we feel that we need many more cases to provide additional mechanistic insights we chose to not diagram any potential mechanism ourselves. As mentioned earlier we are developing the technology to increase scale.

10. Figure 4b – The values for RI and gene targeting should be indicated as in Figure 3.

This has now been done.

11. The references have been separated between references 31 and 32.

This has now been corrected.

Reviewer #3 (Remarks to the Author):

This ms. describes a well-designed series of experiments showing that pol theta play an important role in RI in mouse cells. The experiments take advantage of their experience with mouse cells for gene targeting and show how RI is regulated in these cell lines. These results will be highly cited.

We thank the reviewer for his/her positive evaluation and kind words

REVIEWERS' COMMENTS:

Reviewer #1 (Remarks to the Author):

My concerns were partially addressed. I still believe that the authors cannot exclude that the direct stimulation of HR in PolQ (and Ku) depleted cells leads to increased gene targeting.

The manuscript nicely shows that PolQ and Lig4 are responsible for all random DNA integration in cells. These sets of results are important and significant. However, the authors provide little evidences to connect the increased frequency of accurate targeting by HR to RI suppression in Polq/Lig DKO. I am not convinced that these two phenomena are linked. Cells are transfected with excess amount of target DNA, and therefore, the donor template is never limiting; there should be plenty of DNA to integrate randomly and accurately. I don't really understand why suppression of RI leads to a shift in HR, and the authors provide no explanation as to why these two have to be linked. Importantly, as the authors point out in figure 4A, there is increased HR in Polq /Ku null cells, so the statement that " a potential increase in HR through the lack of PolQ is unlikely to contribute to increase gene targeting efficiency" is not very accurate.

Either provide data that directly links RI suppression to accurate gene targeting by HR, or adjust the statement to take into account that Polq and Ku can act locally at the level of the Cas9 triggered breaks to block HR.

Reviewer #2 (Remarks to the Author):

The authors have addressed all of my major concerns except one (see below). The addition of the CRISPR-induced gene targeting in the various genetic backgrounds nicely complements the IR experiments.

Supplementary Table 2 showing the RI junctions is a good addition but still requires more information. A third line for each junction, showing the plasmid sequence and placed below the genomic sequence, is customary for this type of analysis. Also, are we looking at both the left and right junctions for each event or only one? Finally, an analogous supplementary table showing the corresponding information for the insertions is needed (the Excel file is just too difficult for the average reader to parse).

Other issues:

1. What do the arrow and asterisk mean in the Western blot in Figure 1C?
2. There are still some typos (e.g. Sup. Figure 4 legend (CRSIPR), the reference needs to be inserted in the discussion paragraph).

Point to point response NCOMMS-17-00468A

Reviewer #1 (Remarks to the Author):

My concerns were partially addressed. I still believe that the authors cannot exclude that the direct stimulation of HR in PolQ (and Ku) depleted cells leads to increased gene targeting.

The manuscript nicely shows that PolQ and Lig4 are responsible for all random DNA integration in cells. These sets of results are important and significant. However, the authors provide little evidences to connect the increased frequency of accurate targeting by HR to RI suppression in Polq/Lig DKO. I am not convinced that these two phenomena are linked. Cells are transfected with excess amount of target DNA, and therefore, the donor template is never limiting; there should be plenty of DNA to integrate randomly and accurately. I don't really understand why suppression of RI leads to a shift in HR, and the authors provide no explanation as to why these two have to be linked. Importantly, as the authors point out in figure 4A, there is increased HR in Polq /Ku null cells, so the statement that " a potential increase in HR through the lack of PolQ is unlikely to contribute to increase gene targeting efficiency" is not very accurate.

Either provide data that directly links RI suppression to accurate gene targeting by HR, or adjust the statement to take into account that Polq and Ku can act locally at the level of the Cas9 triggered breaks to block HR.

In his/her argument the reviewer is correct in pointing out that the donor template is not limiting, and that suppression of RI should not necessarily lead to shift in HR, if by shift we meant "switch from one mechanism of integration to another". This however was not what we wanted to say, and we now see which parts of the text were not sufficiently clear.

The Rad54-GFP targeting assay measures relative gene targeting efficiency, i.e. the ratio between targeted and random integrants. We adjusted the text in the third paragraph on page 5 and second paragraph on page 6 to explicitly state what the Rad54-GFP assay measures and clarify that the shift refers to the ratio, and that the change in the ratio was accompanied by reduction in the total number of integrants.

We also, as requested by the reviewer, nuanced the statement about a suppressive effect of EJ proteins on GT, taking into account that we have observed such an effect in the DR-GFP assay.

Finally, we feel that we did not accurately phrase the outcome of the experiment that was included in the resubmission (CRISPR/Cas9 stimulated GT). Instead of "*As most gene targeting strategies currently involve stimulation by DSB induction at the target locus,*

we tested whether loss of Pol θ and C-NHEJ could still enhance CRISPR/Cas9-stimulated recombination and indeed it did (Supplementary Fig. 4).”, it should read “As most gene targeting strategies currently involve stimulation by DSB induction at the target locus, we tested whether loss of Pol θ and C-NHEJ could still be beneficial in the context of CRISPR/Cas9-stimulated gene targeting and indeed it was (Supplementary Fig. 4).”

Reviewer #2 (Remarks to the Author):

The authors have addressed all of my major concerns except one (see below). The addition of the CRISPR-induced gene targeting in the various genetic backgrounds nicely complements the IR experiments.

Supplementary Table 2 showing the RI junctions is a good addition but still requires more information. A third line for each junction, showing the plasmid sequence and placed below the genomic sequence, is customary for this type of analysis. Also, are we looking at both the left and right junctions for each event or only one? Finally, an analogous supplementary table showing the corresponding information for the insertions is needed (the Excel file is just too difficult for the average reader to parse).

Supplementary Table 2 has been modified such that it is consistent with other published work. Also all insertions have been graphically depicted. Finally the Table title was changed to indicate that only the right junction was determined (by TAIL PCR).